# The Emerging Role of IGF2BP2 in Cancer Therapy Resistance: From Molecular Mechanism to Future Potential

**DOI:** 10.3390/ijms252212150

**Published:** 2024-11-12

**Authors:** Die Li, Shiqi Hu, Jiarong Ye, Chaojie Zhai, Jipeng Liu, Zuao Wang, Xinchi Zhou, Leifeng Chen, Fan Zhou

**Affiliations:** 1Department of General Surgery, The Second Affiliated Hospital of Nanchang University, Nanchang 330006, China; wyyxld@126.com (D.L.); hushiqi2828@163.com (S.H.); 17826938900@163.com (J.Y.); 16638044421@163.com (C.Z.); liujipeng@163.com (J.L.); 18772248561@139.com (Z.W.); 16638044421@126.com (X.Z.); 2Department of Oncology, Shanghai Medical College, Fudan University, Shanghai 200433, China; 3Department of Oncology, The Second Affiliated Hospital of Nanchang University, Nanchang 330006, China; 4Medical Center for Cardiovascular Diseases, Neurological Diseases and Tumors of Jiangxi Province, The Second Affiliated Hospital of Nanchang University, Nanchang 330006, China

**Keywords:** IGF2BP2, drug resistance, m6A modification, RNA modifications, cancer

## Abstract

Tumor resistance is one of the primary reasons for cancer treatment failure, significantly limiting the options and efficacy of cancer therapies. Therefore, overcoming resistance has become a critical factor in improving cancer treatment outcomes. IGF2BP2, as a reader of m6A methylation, plays a pivotal role in the post-transcriptional regulation of RNA through the methylation of m6A sites. It not only contributes to cancer initiation and progression but also plays a key role in tumor drug resistance. This review provides a comprehensive summary of the mechanisms by which IGF2BP2 contributes to therapy resistance, with the aim of improving the efficacy of chemotherapy in cancer treatment. Advancing research in this area is crucial for developing more effective therapies that could significantly improve the quality of life for cancer patients.

## 1. Introduction

Cancer is considered a leading cause of death worldwide, claiming nearly 10 million lives in 2020, or approximately one in six deaths. According to the World Health Organization, cancer is a general term that includes a wide range of diseases that can affect any part of the body [1]. RNA modification is a post-transcriptional regulatory mechanism that is widely distributed across various types of RNA, including messenger RNA (mRNA), transfer RNA (tRNA), ribosomal RNA (rRNA), small non-coding RNA, and long non-coding RNA (lncRNA) [2]. Recently, RNA modifications, novel hotspot areas of epigenetic research, have been shown to play crucial roles in protumor and anti-tumor immunity [3]. There are over 170 types of RNA modifications, including N6-methyladenosine (m6A), N6,2′-O-dimethyladenosine (m6Am), 5-methylcytidine (m5C), 5-hydroxylmethylcytidine (hm5 C), and N1-methyladenosine (m1A). Among these, m6A methylation is the most abundant internal mRNA modification in eukaryotes, which are not only in messenger RNAs but also in non-coding RNAs. And m6A sites are enriched in 3′ untranslated regions (UTRs) especially located near stop codons [4].

M6A methylation is the most common form of mRNA modification, referring to the chemical modification where a methyl group (-CH3) is added to the nitrogen atom at the N6 position of adenosine (A) within RNA molecules, particularly in messenger RNA (mRNA). This modification is a type of post-transcriptional change, meaning it occurs after the RNA molecule has been synthesized [5]. M6A methylation is a dynamic and reversible co-transcriptional process, facilitated by “writers” (m6A methyltransferases) that add the m6A modification, “erasers” (demethylases) that remove the m6A modification, and “readers” (m6A-binding proteins) that recognize these modifications and mediate their functions [6]. During mRNA transcription in the nucleus, methyltransferases add a methyl group to the N6 position of adenine through co-transcriptional modification [7]. This modification is subsequently recognized by nuclear methylation readers to exert biological functions. Meanwhile, demethylases also play a role in this process, leading to demethylation at certain sites [8] (Figure 1).

The methylation of m6A sites regulates the post-transcriptional modifications of RNAs in several ways, including splicing, exporting, stabilization, translation, and decay. Recently, it has been demonstrated that m6A can participate in the regulation of biological processes through a variety of mechanisms, thus playing a key role in cancer. The methylation of m6A is regulated by regulators, consisting of “writers”, “erasers”, and “readers”. Writers are methyltransferases, which install m6A methyl groups on RNA, while erasers are demethylases, which remove RNA m6A methylation reversibly [9]. Readers are proteins that perform the biological functions of m6A methylation, and insulin-like growth factor 2 mRNA-binding proteins (IGF2BPs) are newly identified members of this group, which include IGF2BP1, IGF2BP2, and IGF2BP3 [10]. They could participate in promoting cancer cell proliferation, survival, and chemoresistance, inhibiting apoptosis, by binding to m6A sites on different RNA sequences and stabilizing them. What is more, they could also be associated with cancer glycolysis, angiogenesis, and immune responses in the tumor microenvironment (TME) [9].

In this review, we would introduce the role and mechanism of IGF2BP2 in tumorigenesis and tumoral development and discuss its detail function in tumoral chemoresistance.

## 2. Structure and Physiological Function of IGF2BP2

### 2.1. Molecular Structure of IGF2BP2

IGF2BP2/IMP2/VICKZ2 is a member of oncofetal mRNA-binding proteins [11], which share a conserved structure of two RNA recognition motifs (RRMs) at the N-terminus, followed by four K-homology (KH) domains (Figure 2). Their molecular masses are 66 kDa. Although the two RRM domains and four KH domains are also found in other RNA-binding proteins, such as hnRNP A1 and KSRP, the specific combination of these six modular architectures is unique to the IGF2BP family [12]. This configuration, characterized by a similar sequence and spacing between the two RRMs, as well as between KH1/2 and KH3/4, stands out due to the significant divergence in the regions between RRM2 and KH1, and between KH2 and KH3. This structural arrangement has led to the hypothesis that the RNA-binding modules function in a cooperative, pairwise manner. These unique structural features are essential for IGF2BP2’s binding specificity and stability with its target mRNAs [13]. The coordinated interaction between the RRMs and KH domains likely enhances the affinity and selectivity of IGF2BP2 for particular mRNA sequences. Additionally, the divergence between certain domains may provide IGF2BP2 with the flexibility needed to engage with a diverse set of RNA targets, contributing to its regulatory versatility.

The KH domains of IGF2BP2 play a pivotal role in post-transcriptional regulation, particularly in the cytoplasmic trafficking and localization of IGF-II mRNA, as well as in the recognition and binding of specific RNA sequences [14]. The crystal structure of the KH3/4 domains revealed a type 1 KH fold (βααββα) arranged in an anti-parallel pseudo-dimer orientation (Figure 1) [15,16]. Notably, the structural organization of IGF2BP2′s dimer varies significantly depending on whether it is bound to RNA. In the absence of RNA (apo state), the dimer adopts a head-to-tail configuration, with KH34 and KH12 domains positioned at the interface. In contrast, when RNA-bound, the dimer exhibits a more symmetric arrangement stabilized by a KH12-KH12 interface [17]. This structural flexibility is essential for the recognition and binding of m6A-modified RNA targets [18]. The KH3/4 domains were shown to specifically bind to AGGU and UGGA motifs in SELEX experiments [15]. Beyond mRNA interactions, IGF2BP2 also binds non-coding RNAs (ncRNAs) via the KH3/4 domains. For example, it interacts with the “CAUCAU” m6A motif at the exon 5–exon 4 junction of circNSUN2, promoting colorectal liver metastasis [19]. However, the stabilizing capacity of the KH3/4 domains is somewhat limited, as their ability to bind additional RNA molecules diminishes significantly once engaged with a target RNA. Interestingly, the structure and function of the KH1/2 domains in IGF2BP2 remain less well characterized. Given the high sequence identity (>70%) between IGF2BP2 and IGF2BP1 in their RNA-binding domains, insights into the KH1/2 domains of IGF2BP2 may be gleaned from studies on IGF2BP1 [15]. In IGF2BP1, KH2 shows lower specificity and more rapid kinetics in RNA binding compared to KH4, suggesting that KH2 may play an early role in RNA binding by facilitating dynamic, non-specific interactions, likely driven by avidity effects. When the KH1 target sequence is present, the encounter complex formed between KH2 and RNA can be stabilized by KH1, contributing to RNA stabilization [16]. In summary, the KH domain architecture enables IGF2BP2 to participate in complex RNA interactions, enhancing its ability to modulate RNA metabolism, including looping and stabilization, which is crucial for the spatial and temporal regulation of gene expression. This regulatory capacity underscores the importance of KH domains in controlling cellular processes, and their dysregulation could have profound implications for diseases such as cancer, where IGF2BP2 plays a role in oncogenic mRNA stabilization.

The RRM domain. In vitro studies have shown that the KH domains primarily govern RNA binding, while the RRM domains play a crucial role in stabilizing the IGF2BP2-RNA complexes, effectively extending the half-life of these interactions [20]. This division of labor between the domains highlights a cooperative mechanism, where the KH domains ensure the specificity and initial binding to the target RNA, and the RRM domains provide the necessary stabilization to maintain the interaction over time.

### 2.2. Physiological Role of IGF2BP2

Unlike IGF2BP1 and IGF2BP3, IGF2BP2 is ubiquitously expressed in normal adult tissues [21], and plays an important role in physiological processes, which include enhancing RNA stability and facilitating RNA translation.

IGF2BP2 maintains RNA stability. IGF2BP2 plays a critical role in maintaining RNA stability, particularly through an m6A-dependent mechanism: (1) Splicing and decapping: m6A modification regulates the splicing and decapping of mRNA, thereby promoting the formation of mature mRNA. Proper splicing and decapping can improve the stability of mRNA [22]. (2) Binding specific proteins: m6A modifications can recruit specific RNA-binding proteins, such as YTHDF family proteins, that protect RNA from degradation or promote its translation by binding to m6A-modified sites [23]. (3) Inhibition of RNA degradation: m6A modification inhibits specific RNA degradation pathways, such as the “deadenylation” pathway, thereby extending the half-life of mRNA. Modified RNA is less likely to be degraded by ribonucleases [24]. (4) m6A methylation can improve the translation efficiency of mRNA and enhance protein synthesis, thereby indirectly improving RNA stability, because more translation products will promote the maintenance of RNA presence in cells [25]. (5) m6A can synergize with other RNA modifications, such as m5C, to form a complex regulatory network that affects RNA stability and function [26]. Some experiments showed that knockdown or knockout of the m6A writer complex resulted in reduced stability of m6A-modified RNAs. Conversely, knockdown or knockout of m6A eraser proteins led to increased stability of specific m6A-containing RNAs. Biochemical studies further confirmed that IGF2BPs, including IGF2BP2, stabilize m6A-modified RNAs across various cellular, physiological, and pathological settings, including human cancer cells such as hepatocellular carcinoma (HepG2), cervical cancer (HeLa), prostate cancer (22Rv1), and acute myeloid leukemia. These findings extend to postnatal liver development and early embryonic stages in mouse oocytes [27,28]. Collectively, these data underscore the role of IGF2BP2 as a key effector in the m6A regulatory pathway. By binding to m6A-modified RNAs, IGF2BP2 ensures their prolonged stability, which is crucial for the sustained expression of genes involved in processes such as cell proliferation and differentiation. In cancer, this stabilization mechanism can be particularly detrimental, as it supports the continued expression of oncogenes and other factors that drive tumor progression and metastasis. However, m6A modification could also be recognized by YTH domain family 2 (YTHDF2), which is another member of the m6A reader, to regulate mRNA degradation [29].

IGF2BP2 facilitates RNA translation. IGF2BP2 also plays a significant role in facilitating RNA translation, in addition to enhancing RNA stability. IGF2BPs promote the translation of both coding and non-coding RNAs [30]. For example, IGF2BP2 can recruit ELAV-like RNA-binding protein 1 (ELAVL1, also known as HuR) to protect m6A-modified mRNAs from degradation, ensuring their translation [18]. This recruitment mechanism exemplifies how IGF2BP2 can enhance the translation of m6A-containing mRNAs, thereby contributing to gene expression regulation at multiple levels. Moreover, IGF2BP2, through its homolog IMP2, has been implicated in controlling the translation of laminin-β2 (LAMB2), a critical component of the extracellular glycoproteins that contribute to the structure and function of the glomerular basement membrane (GBM). Dysregulation of this process can lead to nephrotic syndrome and progressive kidney dysfunction due to the impaired development of the GBM [31].

IGF2BP2 could influence physiological metabolic processes. IGF2BP2 plays a pivotal role in influencing various physiological metabolic processes. It has been shown to be involved in adipocyte development, particularly in the early differentiation of adipocyte-derived stem cells (ADSCs) into preadipocytes. Studies have demonstrated that mice with Imp2 deletion in mesenchymal stem cells (MSCs) exhibit resistance to diet-induced obesity without impairments in glucose and insulin tolerance, ultimately protecting against the development of type 2 diabetes (T2D). Mechanistically, IGF2BP2 binds to Wnt receptor Fzd8 mRNA and promotes its degradation by recruiting the CCR4-NOT deadenylase complex in an mTOR-dependent manner, thereby maintaining white adipose tissue homeostasis by controlling mRNA stability in ADSCs [32]. In addition, IGF2BP2 and Mettl3 jointly regulate the stability of mRNAs coding for key glycolytic enzymes in beige adipocytes, impacting both glycolysis and thermogenesis [33]. This regulation is essential for energy balance and metabolic function in adipose tissue. Furthermore, the deletion of IMP2 in pancreatic β-cells leads to impaired compensatory β-cell proliferation and function. Mechanistically, IMP2 directly binds to Pdx1 mRNA, promoting its translation in an m6A-dependent manner, highlighting its role as a crucial regulator of pancreatic β-cell proliferation and function [34].

IGF2BP2 could influence embryonic development. IGF2BP2 plays a critical role in embryonic development, particularly in regulating key cellular functions. It has been shown to significantly affect retinal pigment epithelium (RPE) phagocytosis by modulating the mRNA stability of transcription factors PAX6 and OTX2. Loss of IGF2BP2 in RPE cells induces inflammatory and aging-like phenotypes, impairing RPE function and leading to retinal dysfunction in vivo [35]. In another context, the microRNA lethal (let)-7e-5p was found to be overexpressed in androgen-treated orchidectomized (ORX) mice compared to ORX controls. The let-7 family is well known for its association with muscle atrophy. Specifically, let-7e-5p downregulates IGF2BP2 expression in myotube cells, inhibiting myosin heavy chain growth and contributing to muscle atrophy [36]. Furthermore, the Lin28b/Hmga2 pathway, which governs tissue development, undergoes postnatal downregulation, limiting the self-renewal capacity of adult hematopoietic stem cells (HSCs) compared to fetal HSCs. IGF2BP2, acting downstream of Lin28b/Hmga2, plays a crucial role in regulating the stability and translation of messenger RNAs necessary for tissue development [37]. These findings highlight IGF2BP2 as a key regulator in multiple developmental processes, from retinal function to muscle growth and hematopoietic stem cell renewal. Dysregulation of IGF2BP2 could have broad implications, potentially contributing to developmental abnormalities, degenerative diseases, and tissue-specific dysfunctions.

## 3. Pathological Function of IGF2BP2 in Tumors

IGF2BP2 plays a critical role in cancer development, progression, and drug resistance by modulating m6A RNA modifications [37,38]. As a reversible and prevalent modification, m6A occurs on RNA molecules (including mRNA and ncRNA) through methylation at the nitrogen-6 position of adenosine within an RRACH sequence (where R = G or A, and H = A, U, or C) [39]. This modification process is coordinated by three types of proteins: writers (methyltransferases), readers (m6A-binding proteins), and erasers (demethylases) [40]. Writers catalyze the addition of methyl groups to target RNAs, while erasers remove these methyl groups, enabling the dynamic regulation of m6A levels. Readers recognize m6A-marked binding sites on RNA, which recruits writers and affects RNA folding, stability, degradation, and cellular interactions [41]. Through these mechanisms, m6A modifications are involved in RNA splicing, translation, export, and decay [42].

In cancer, IGF2BP2, a common m6A reader, exhibits aberrant expression, impacting the stability and expression of various mRNAs associated with oncogenes (e.g., MYC [43], KRAS [44], BCL2 [45]), metabolism-related genes (e.g., GLUT1 [46], HK2 [47]), stemness-related genes (e.g., SOX2 [48], NANOG [49], OCT4 [50]), and drug-resistance genes (e.g., ABCB1 [51], which encodes multidrug resistance proteins). Consequently, IGF2BP2 significantly influences tumor growth, progression, and resistance to therapies.

### 3.1. IGF2BP2 Expression in Pan-Cancer

IGF2BP2 exhibited differential expression across various cancer types, including Breast Invasive Carcinoma (BRCA), Cholangiocarcinoma (CHOL), Colon Adenocarcinoma (COAD), Esophageal Carcinoma (ESCA), GBM, head and neck squamous cell carcinoma (HNSC), human papilloma virus-negative head and neck squamous cell carcinoma (HNSC-HPV), Kidney Chromophobe (KICH), kidney renal clear cell carcinoma (KIRC), Kidney Renal Papillary Cell Carcinoma (KIRP), liver hepatocellular carcinoma (LIHC), Lung Squamous Cell Carcinoma (LUSC), Pheochromocytoma and Paraganglioma (PCPG), Prostate Adenocarcinoma (PRAD), Stomach Adenocarcinoma (STAD), and thyroid carcinoma (THCA) when compared to normal tissue. In several cancer types, including CHOL, COAD, ESCA, GBM, HNSC, KICH, KIRP, LIHC, LUSC, STAD, and THCA, IGF2BP2 was found to be overexpressed in tumor cells. However, in BRCA, KIRC, PCPG, and PRAD, its expression was lower in tumor cells compared to normal tissue. Additionally, IGF2BP2 was overexpressed in HPV-negative HNSC and in metastatic Skin Cutaneous Melanoma (SKCM); however, this analysis is based solely on existing database data, and research on actual prognosis remains lacking. These findings highlight the differential roles of IGF2BP2 in tumor biology, with its expression patterns potentially serving as biomarkers for cancer progression and therapeutic targeting in various malignancies (Appendix A). A comprehensive pan-cancer analysis has demonstrated that elevated expression levels of IGF2BP2 are associated with poor prognosis across various cancer types. Specifically, high IGF2BP2 expression significantly correlates with unfavorable outcomes in patients with adrenocortical carcinoma (ACC) (HR = 1.17, *p* < 0.03), bladder carcinoma (BLCA) (HR = 1.09, *p* < 0.01), kidney renal clear cell carcinoma (KIRC) (HR = 1.21, *p* < 0.001), low-grade glioma (LGG) (HR = 1.48, *p* < 0.001), head and neck squamous cell carcinoma (HNSC) (HR = 1.17, *p* < 0.001), lung adenocarcinoma (LUAD) (HR = 1.12, *p* < 0.02), acute myeloid leukemia (LAML) (HR = 1.10, *p* < 0.03), liver hepatocellular carcinoma (LIHC) (HR = 1.11, *p* < 0.01), pancreatic adenocarcinoma (PAAD) (HR = 1.45, *p* < 0.001), and mesothelioma (MESO) (HR = 1.27, *p* < 0.001). These findings suggest that IGF2BP2 may act as an oncogene, contributing to adverse prognoses in multiple cancer types [52]. In various preclinical models of colorectal cancer (CRC), IGF2BP2 was identified as the most prevalent IGF2BP family member in both primary and metastatic CRC, showing a correlation with tumor stage in patient samples and promoting tumor growth [53]. Furthermore, elevated IGF2BP2 expression in primary tumor tissue was significantly linked to resistance against multiple therapies, including selumetinib, gefitinib, and regorafenib in patient-derived organoids (PDOs), as well as 5-fluorouracil and oxaliplatin [53]. Similar associations have been observed in other cancer types, such as breast cancer [51], lung cancer [54], ovarian cancer [55], GBM [56], etc. Notably, IGF2BP2 may also play a protective role in cancers like clear cell renal cell carcinoma (ccRCC) [57]. As previously noted, IGF2BP2 functions as an m6A RNA reader, primarily exerting its effects through RNA modification. In the following sections, we will review the role of IGF2BP2 in modulating tumor RNA.

### 3.2. IGF2BP2 Function in Tumor RNA

Recent studies have demonstrated that m6A modifications play a critical role in cancer biology [58]. Disrupting the expression or activity of IGF2BP2 may counteract tumorigenesis driven by aberrant m6A processes. Structural data suggest that IGF2BP2 recognizes RNA structural changes induced by the so-called “m6A-switch” [59]. In this context, m6A alters local RNA structures, facilitating the binding of RNA-binding proteins such as hnRNPC and hnRNP A2B1 [60,61]. Evidence indicates that IGF2BP2 regulates gene expression by binding to m6A-modified sites on target mRNAs, thereby influencing various stages of RNA metabolism. This regulation affects multiple oncogenic processes, including maintaining cancer stem cell stemness, promoting tumor cell proliferation, enhancing migration, stimulating glycolysis, driving cell cycle transitions, and supporting angiogenesis. In the following sections, we will first introduce IGF2BP2’s function in tumor RNA regulation.

IGF2BP2 enhances tumor RNA stability. IGF2BP2 can stabilize tumor RNA through an m6A-dependent mechanism, similar to its role in physiological processes. These RNAs include both coding RNAs and non-coding RNAs (ncRNAs). NcRNAs, such as long non-coding RNAs (lncRNAs), circular RNAs (circRNAs), and microRNAs (miRNAs), are critical regulators of tumor initiation and progression [62]. IGF2BP2 relies on its characteristic KH domains to recognize m6A signals on RNAs, thereby maintaining their stability [63]. For instance, IGF2BP2 has been confirmed to bind MSX1 and JARID2 via its KH3-4 domain in CRC [64]. Additionally, the KH3-4 domain of IGF2BP2 interacts specifically with the “CAUCAU” motif at the exon 5–exon 4 junction of circNSUN2, forming an RNA–protein ternary complex that induces CRC metastasis [19]. IGF2BP2 also participates in m6A processes mediated by abnormally expressed m6A “erasers”. For example, the depletion of FTO increases IGF2BP2’s binding affinity for metastasis-associated protein 1 (MTA1) mRNA, contributing to CRC metastasis [65]. However, further experimental studies are required to fully elucidate the precise mechanisms through which IGF2BP2 stabilizes RNA.

IGF2BP2 facilitates tumor RNA translation. In addition to enhancing RNA stability, IGF2BP2 also promotes RNA translation [30]. For example, in papillary thyroid cancer, IGF2BP2 increases the translational efficiency of erb-b2 receptor tyrosine kinase 2 (ERBB2) by binding to m6A motifs in its coding sequence (CDS), thereby conferring resistance to tyrosine kinase inhibitors [66]. Similarly, the translational regulation of pancreatic and duodenal homeobox 1 (PDX1) in pancreatic β cells and ERBB2 in radioiodine-refractory papillary thyroid cancer also relies on IGF2BP2’s recognition of m6A methylation [66]. In osteosarcoma, IGF2BP2 interacts with m6A motifs within the CDS of MN1, promoting both mRNA stability and translation [67]. Moreover, in human embryonic rhabdomyosarcoma, IGF2BP2 phosphorylation by mTOR enhances its translational activity, regulating the initiation of IGF2 leader 3 mRNA translation via EIF4E and activating the 5′ cap-independent internal ribosome entry site (IRES) [68]. These findings strongly suggest that IGF2BP2 regulates mRNA translation through m6A methylation, highlighting a novel oncogenic mechanism mediated by IGF2BP2 and positioning it as a potential therapeutic target in cancer. Additionally, IGF2BP2 influences mRNA through other mechanisms. In glioblastoma, IGF2BP2 binds to oxidative phosphorylation-related mRNAs, such as NDUFS3 and COX7b, delivering them to mitochondrial polysomes for translation [69]. These insights underscore the functional diversity of IGF2BP2 in mRNA processing.

Various factors affect the binding ability of IGF2BP2 and RNA. IGF2BP2’s ability to bind different RNAs is crucial for its role in cancer progression, as this binding influences a wide range of oncogenic processes. The RNA-binding capacity of IGF2BP2 can be modulated by various factors, including proteins, mRNAs, and non-coding RNAs (ncRNAs). For example, in breast cancer, the oncogene Aurora kinase A (AURKA) enhances IGF2BP2’s binding to m6A-modified transcripts without promoting its nuclear translocation [70]. Additionally, in myeloid leukemia, YBX1 plays a key role in facilitating IGF2BP2’s recognition of m6A-modified RNAs [71]. These findings emphasize the complex regulation of IGF2BP2’s RNA-binding activity, which is critical for its oncogenic functions.

Although most ncRNAs are not translated into proteins, they play significant roles in gene regulation, particularly in cancer development and progression. Some ncRNAs can enhance RNA binding processes that promote cancer progression, while others can competitively bind to the KH domains of IGF2BP2. This competitive binding interferes with IGF2BP2’s recognition of m6A-modified RNAs, thereby inhibiting their expression and suppressing cancer progression [72,73]. These findings suggest that the effects of ncRNAs on the RNA-binding capacity of IGF2BP2 are complex. Therefore, further in-depth investigation into this mechanism is warranted, as it could provide a more comprehensive understanding of the role of IGF2BP2 in cancer. Identifying molecules that can specifically inhibit IGF2BP2’s RNA-binding ability could offer more effective therapeutic strategies for treating cancers. Long non-coding RNAs (lncRNAs), a subgroup of ncRNAs longer than 200 nucleotides, play significant roles in various biological processes and diseases, including cancer. In cancer, lncRNAs have been shown to interact with m6A methylation, binding to IGF2BP2 to stabilize themselves [74]. A notable example is the lncRNA DANCR, which is regulated by IGF2BP2 in an m6A methylation-dependent manner, promoting stem cell-like properties, cancer cell proliferation, and contributing to the progression of prostate cancer (PC) [49]. Recent studies have revealed a crosstalk between IGF2BP2 and the lncRNA ZFAS1, which promotes ATP hydrolysis and the Warburg effect, playing a critical role in mitochondrial energy metabolism in colon cancer [75]. Additionally, IGF2BP2 enhances the expression of the lncRNA TRPC7-AS1, which in turn upregulates HMGA2 expression, driving hepatocellular carcinoma progression [76].

One of the key roles of miRNAs in cancer is to mediate mRNA silencing [77]. As previously mentioned, IGF2BP2 can disrupt miRNA-dependent mRNA decay by binding to miRNA target sites, thereby weakening the interaction between miRNAs and Ago2 or acting as cytoplasmic “safe houses” to protect mRNAs in the cancer context [78,79]. Activation of the IGF2BP2 family has been shown to promote oncogenic transformation through its interaction with Dicer, forming part of an oncogenic network that involves miRNAs and complex post-transcriptional regulatory mechanisms [80]. Notably, IGF2BP2 protects the target genes of the let-7 miRNA family from silencing, helping to maintain glioblastoma stem cell populations [81].

Research on how IGF2BP2 regulates circular RNAs (circRNAs) in cancer is still limited, with only a few studies indicating that IGF2BP2 functions by recognizing and binding to m6A-modified sites in circRNAs. For example, in cervical cancer, IGF2BP2 recognizes m6A modifications in circARHGAP12, which in turn promotes the stabilization of forkhead box M1 (FOXM1) mRNA [82]. Additionally, the hsa_circ_0003258–IGF2BP3–HDAC4 complex has been shown to enhance the stability of histone deacetylase 4 (HDAC4) mRNA, as both hsa_circ_0003258 and HDAC4 contain m6A modification sites [73]. Moreover, the depletion of circCD44 or IGF2BP2 affects the levels of m6A-modified c-MYC mRNA, suggesting that the interaction between circCD44 and IGF2BP2 may stabilize c-MYC in triple-negative breast cancer (TNBC) [44]. Based on these findings, it is reasonable to speculate that IGF2BPs can bind to m6A-modified circRNAs, regulating the expression of downstream genes. However, research also suggests that IGF2BPs interact with circRNAs through mechanisms beyond m6A recognition. For instance, the KH3-4 di-domain of IGF2BP2 is essential for its interaction with circNSUN2 and HMGA2, allowing IGF2BP2 to bind to the CAUCAU motif of circNSUN2 and enhance HMGA2 mRNA stability. Interestingly, this interaction stabilizes HMGA2 independently of m6A methylation [19].

### 3.3. Role of IGF2BP2 in Tumor Development

Cancer stem cell self-renewal. Self-renewal refers to the process by which stem cells divide to maintain a sufficient population throughout life. Cancer stem cells (CSCs), characterized by their ability to self-renew and form clones, are strongly linked to cancer recurrence, metastasis, and therapy resistance. Therefore, understanding the mechanisms that sustain CSC stemness is crucial for identifying therapeutic targets in cancer. IGF2BP2 has been identified as a key regulator of stem-like, tumorigenic properties in hepatocellular carcinoma (HCC), glioblastoma, osteosarcoma, CRC, acute myeloid leukemia (AML), and other cancers [81,83]. In CRC, IGF2BP2 reinforces stem cell self-renewal by recognizing the coding sequence (CDS) of the SRY-box transcription factor 2 (SOX2), thereby preventing its degradation [48]. Additionally, IGF2BP2 stabilizes PRMT6 mRNA in an m6A-dependent manner, which in turn catalyzes H3R2me2a, leading to the suppression of the lipid transporter MFSD2A. Loss of PRMT6 results in the upregulation of MFSD2A, increasing docosahexaenoic acid (DHA) levels, impairing leukemia stem cell (LSC) maintenance, and promoting AML development [84]. Furthermore, AURKA enhances IGF2BP2’s binding to m6A-modified DROSHA mRNA, stabilizing DROSHA transcripts. This interaction is further strengthened by the direct binding of AURKA to DROSHA, promoting stem cell-like properties in breast cancer [70].

Tumor proliferation. IGF2BP2 plays a critical role in promoting tumor cell proliferation by regulating the cell cycle. For example, in triple-negative breast cancer (TNBC), IGF2BP2 enhances cell proliferation and facilitates the G1/S phase transition by directly regulating CDK6 in an m6A-dependent manner and recruiting EIF4A1 to promote CDK6 translation output [85]. In CRC, IGF2BP2 binds to the coding sequence (CDS) of SOX2, promoting tumor progression through a mechanism dependent on METTL3 [48]. Head and neck squamous cell carcinoma (HNSCC) is highly aggressive and prone to cervical lymph node metastasis, leading to poor prognosis. In HNSCC, IGF2BP2 has emerged as a key oncogene driven by oncogenic super-enhancers (SEs), which maintain its overexpression and cancer progression. The SE region of IGF2BP2 is enriched with H3K27Ac, BRD4, and MED1 markers, which, paradoxically, inhibit its transcription by deactivating the SE-associated transcriptional program. Additionally, KLF7 enhances IGF2BP2 transcription by binding directly to its promoter and SE regions. A positive correlation between KLF7 and IGF2BP2 levels has been observed in HNSCC tissues, with high expression levels of both genes being associated with poor patient outcomes, making them potential prognostic markers and therapeutic targets in HNSCC [86]. Moreover, m6A modifications and SEs are heavily involved in the development of AML. In AML, IGF2BP2 stabilizes DDX21 mRNA in an m6A-dependent manner, contributing to poor patient survival [87].

Tumor metastasis. Metastasis is the deadliest aspect of cancer, and IGF2BP2 has been shown to promote tumor metastasis by regulating processes such as tumor angiogenesis and epithelial–mesenchymal transition (EMT). In terms of angiogenesis, IGF2BP2 modulates the expression of ephrin type-A receptor 2 (EphA2) and vascular endothelial growth factor A (VEGFA) to facilitate vasculogenic mimicry in CRC [88]. Beyond VEGFA, IGF2BP2 can also indirectly regulate hypoxia-inducible factor 1 subunit alpha (HIF1A) and matrix metalloproteinase 14 (MMP14) to drive vasculogenic mimicry in glioma [89]. In LUAD, IGF2BP2 enhances the RNA stability of FLT4 via m6A modification, activating the PI3K-Akt signaling pathway, which leads to increased angiogenesis, metastasis, and poor clinical outcomes [90]. In gallbladder cancer (GBC), the long non-coding RNA TRPM2-AS is upregulated and closely associated with angiogenesis and poor prognosis. The high expression and stability of TRPM2-AS are attributed to m6A modification, which is recognized by IGF2BP2. TRPM2-AS, when loaded into exosomes, is transported from GBC cells to human umbilical vein endothelial cells (HUVECs), where it suppresses NUMB expression through PABPC1, ultimately activating the pro-angiogenic NOTCH1 signaling pathway [91]. Furthermore, cancer cells release extracellular vesicles (EVs) that influence macrophage polarization toward tumor-associated macrophages (TAMs), which in turn promote cancer metastasis [91]. IGF2BP2 plays a key role in determining the cargo of EVs released by cancer cells, thereby modulating their effects on macrophages and promoting cancer cell migration.

Evading programmed cell death. IGF2BP2 plays a key role in dysregulating programmed cell death, particularly in ferroptosis and autophagy, within cancer cells. Apoptosis is a form of programmed cell death responsible for eliminating stressed, damaged, malignant, or infected cells [92]. Ferroptosis, a recently discovered iron-dependent form of cell death, is also affected by IGF2BP2. In hypopharyngeal squamous cell carcinoma, IGF2BP2, through m6A modification, stabilizes NFE2L2/NRF2 mRNA, resulting in resistance to ferroptosis [93]. Autophagy, a cellular process essential for meeting metabolic needs and renewing organelles to maintain homeostasis, plays a complex, dual role in cancer [94]. Recent evidence has revealed that IGF2BPs are involved in regulating autophagy during tumor progression. For instance, in clear cell renal cell carcinoma (ccRCC), IGF2BP2 stabilizes salt-inducible kinase 2 (SIK2) mRNA via FTO-mediated m6A modification, enhancing autophagic flux and reducing ccRCC growth and metastasis [95]. On the contrary, in gastrointestinal stromal tumors, IGF2BP2 stabilizes the deubiquitinase ubiquitin-specific peptidase 13 (USP13), which extends the half-life of autophagy-related protein 5 (ATG5). This promotes pro-survival autophagy and contributes to drug resistance [96]. Regardless of the specific role autophagy plays in different cancers, IGF2BP2 consistently functions as an oncogene. Programmed cell death mechanisms are crucial in determining resistance to radiotherapy and chemotherapy. Therefore, targeting IGF2BP2 could improve the sensitivity of tumors to these treatments, potentially overcoming resistance and enhancing the effectiveness of anti-tumor therapies.

Metabolic reprogramming. Metabolic reprogramming, particularly in glucose, fatty acid, and amino acid metabolism, allows cancer cells to alter their metabolic pathways to meet energy demands, thus promoting proliferation and growth. This process not only enables cancer cells to resist external stress but also grants them new functional capacities, frequently contributing to tumorigenesis. Notably, IGF2BP2 has been shown to influence metabolic reprogramming. The Warburg effect, or aerobic glycolysis, is a common metabolic reprogramming pathway in tumors. IGF2BP2 recognizes and upregulates m6A-modified Apolipoprotein E (APOE), promoting glycolysis and tumor growth in papillary thyroid cancer [97]. Additionally, IGF2BP2 stabilizes hexokinase 2 (HK2), and both IGF2BP2 and IGF2BP3 stabilize solute carrier family 2 member 1 (SLC2A1, GLUT1), which activate glycolysis to promote CRC progression [98]. Beyond mRNAs regulated by IGF2BP2, non-coding RNAs (ncRNAs) also play a role in regulating glucose metabolism in cancer cells. For example, IGF2BP2 promotes the Warburg effect by stabilizing the ZFAS1/Obg-like ATPase 1 (OLA1) axis in CRC [75]. Furthermore, glutamine, an anaplerotic substrate that replenishes the tricarboxylic acid cycle, supports tumor proliferation. Remarkably, IGF2BP2 regulates glutamine uptake and metabolism by upregulating MYC, glutamic–pyruvic transaminase 2 (GPT2), and solute carrier family 1 member 5 (SLC1A5) in AML, demonstrating that IGF2BP2s can influence amino acid metabolism in cancer [99]. These studies underscore that IGF2BP2 plays a pivotal role in the metabolic reprogramming of cancer cells through m6A-dependent mechanisms. Targeting IGF2BP2 could disrupt the metabolic state of tumor cells and the tumor microenvironment, thereby inhibiting tumor growth.

TME. We propose that the role of IGF2BPs in the TME is multifaceted, influencing not only cancer cell biology but also their interactions with immune cells, prompting further investigation into IGF2BPs’ role in the immunosuppressive TME. The TME is a complex ecosystem in which cancer cells thrive, encompassing various components such as cancer-associated cells, the extracellular matrix, and cytokines. Due to metabolic dysregulation in cancer cells and other contributing factors, an immunosuppressive TME often develops, enabling cancer cells to evade immune surveillance—a key factor in the limited effectiveness of current cancer therapies. IGF2BP2 has been shown to promote the activity of immunosuppressive cells within the TME. These immunosuppressive cells include TAMs, myeloid-derived suppressor cells (MDSCs), regulatory T cells (Tregs), and carcinoma-associated fibroblasts (CAFs). TAMs arise from the infiltration of peripheral immune cells into tumor tissues and play a crucial role in mediating immune escape within the immunosuppressive TME. In CRC patients, IGF2BP2 has been implicated in the immune response of peripheral blood immune cells. Moreover, a ternary complex consisting of circITGB6, IGF2BP2, and FGF9 has been shown to promote TAM polarization to the M2 phenotype, which is associated with tumor-promoting activities in ovarian cancer [100,101].

Other tumor-associated processes. A review of previous studies reveals that, while some research has shown IGF2BP2’s ability to inhibit cancer progression, the majority of evidence supports its role in promoting cancer. IGF2BP2 has been implicated in suppressing anti-tumor immune responses, enhancing the function of immunosuppressive cells, adapting to hypoxic conditions, and driving cancer metabolism, angiogenesis, drug resistance, metastasis, and cell cycle progression. For instance, IGF2BP2 has been shown to increase the stability of SLUG mRNA, a key transcription factor involved in EMT, by binding to its m6A site within the coding sequence (CDS). This stabilization promotes lymphatic metastasis in head and neck squamous cell carcinoma (HNSCC) [73]. IGF2BPs facilitate cancer cells’ adaptation to hypoxia, a hallmark of the TME caused by rapid tumor growth. Hypoxia leads to increased expression of hypoxia-inducible factors (HIFs), which in turn promote cancer immune evasion. Additionally, hypoxia-induced downregulation of FTO has been shown to enhance metastasis in CRC through an m6A-IGF2BP2-dependent mechanism [65]. Notably, the binding of the lncRNA HIF1A-AS2 to IGF2BP2 supports the adaptation of glioblastoma multiforme stem cells to hypoxia [102]. This adaptive response helps prevent hypoxia-induced necrosis of cancer cells and contributes to the formation of an immunosuppressive TME.

Collectively, these studies demonstrate that IGF2BP2 plays a pivotal role in promoting tumor angiogenesis and EMT, both of which are critical for tumor metastasis. And IGF2BP2-mediated m6A interactions across different cell types are summarized in Table 1. So, targeting and inhibiting IGF2BP2 expression could therefore suppress tumor metastasis, offering a promising strategy for slowing cancer progression.

## 4. Impact of IGF2BP2 on the Cancer Treatment

Although targeted therapies and chemotherapy have demonstrated considerable clinical efficacy, the emergence of drug resistance plays a pivotal factor in treatment failure [110]. Recent investigations have elucidated that the RNA-binding protein IGF2BP2 exerts a significant role in influencing cancer treatment outcomes by regulating the resistance of cancer treatment (Table 2).

### 4.1. Chemotherapy Resistance

Chemotherapeutic drugs are commonly used in clinical practice. Based on their mechanisms of action and chemical structures, they can be divided into platinum-based drugs, alkylating agents, antimicrotubular agents, anti-tumor antibiotics, vitamin derivatives, and so on [119]. The role of IGF2BP2 in the resistance to different chemotherapy drugs will be described in the following (Figure 3).

Platinum-based drugs. Studies indicate that the expression of IGF2BP2 in primary tumor tissues is significantly correlated with the resistance of patient-derived organoids (PDOs) and patient-derived xenografts (PDXs) to various chemotherapeutic agents, including cisplatin (CDDP), oxaliplatin, and other targeted drugs [53]. This suggests that IGF2BP2 may influence the resistance of cancer cells to platinum-based drugs by regulating genes associated with drug resistance.

Platinum-based drugs are widely used for the chemotherapy of various solid tumors, but resistance remains a major clinical challenge [120]. IGF2BP2 interacts with non-coding RNAs (ncRNAs) to modulate the expression of drug resistance-related genes, thereby affecting the chemotherapy response of cancer cells [121]. For instance, circular RNAs (circRNAs) play crucial roles in cancer proliferation, metastasis, and chemotherapy resistance [111]. Research shows that IGF2BP2 binds to the CAUC motif of circITGB6 via its KH1-2 domains, enhancing the stability of FGF9 mRNA, which in turn promotes the polarization of macrophages to the M2 phenotype in the TME and induces cisplatin resistance in ovarian cancer cells [100]. Additionally, circPBX3 interacts with IGF2BP2 to increase the stability of ATP7A mRNA, thereby elevating ATP7A protein levels and contributing to the efflux of cisplatin from cancer cells, further promoting resistance [55]. Moreover, microRNA (miRNA) is another way to regulate the biological function of IGF2BP2. For example, miR-96-5p enhances the expression of IGF2BP2, increasing cervical cancer cell resistance to cisplatin [112].

A long non-coding RNA associated with oxaliplatin resistance in NASH-HCC (lnc-OXAR) was identified and upregulated in patients with oxaliplatin resistance. It enhances Ku70 stability, protecting cancer cells from double-strand DNA breaks and playing a crucial role in oxaliplatin resistance. m6A methylation modification upstream of lnc-OXAR can prevent its degradation through an IGF2BP2-dependent pathway mediated by WTAP, thus maintaining its stability and promoting oxaliplatin resistance [113]. These studies demonstrated that IGF2BP2 significantly influences cisplatin resistance by regulating genes associated with platinum-based drug resistance. Targeting IGF2BP2 and its related ncRNAs may offer new therapeutic strategies for overcoming cisplatin resistance, enhancing the response of patients to chemotherapy and improving clinical outcomes.

Temozolomide (TMZ), an important member of the alkylating agents, is primarily used to treat GBM. It exerts its anti-cancer effect by specifically alkylating DNA, thereby disrupting tumor cell proliferation [18]. Its favorable oral bioavailability and relatively low toxicity have established its significance in clinical applications [122]. However, the development of resistance to TMZ remains a major challenge in treatment. The role of IGF2BP2 in TMZ resistance is of considerable importance.

Analysis of TCGA-LGG samples revealed a significant association between the high expression of IGF2BP2 and resistance markers. This finding suggests that IGF2BP2 may influence the efficacy of TMZ by modulating resistance-related pathways [123]. Cell Counting Kit-8 (CCK-8) assays have validated that the knockdown of IGF2BP2 increases cellular sensitivity to temozolomide treatment. As for the underlying mechanism, firstly, IGF2BP2 was reported to regulate insulin-like growth factor 2 (IGF2) mRNA translation, which activated the phosphoinositide 3-kinase (PI3K)/Akt signaling pathway, thus enhancing the survival capacity of tumor cells and increasing the resistance to TMZ [114]. Secondly, IGF2BP2 can stabilize cancer-progressed genes, thus promoting drug resistance. For instance, IGF2BP2 is closely related to the stabilization of SOX2. The complex formation between IGF2BP2 and DHX9 mediated by GSCAR stabilizes SOX2 mRNA, leading to its elevated expression. The high expression of SOX2 enhances the self-renewal capacity of glioma cells while also increasing their resistance to TMZ [115]. In addition, HOXD-AS2 (HOXD cluster antisense RNA 2) is an lncRNA transcript located at the HOX gene locus. Low levels of HOXD-AS2 have been found to render glioblastoma multiforme more sensitive to temozolomide. HOXD-AS2 can form a complex with IGF2BP2 and upregulate the STAT3 signaling pathway, thereby creating a positive feedback loop that regulates the sensitivity of glioblastoma multiforme to TMZ [28].

Paclitaxel. Paclitaxel belongs to antimicrotubular agents and is widely used in clinical practice. IGF2BP2 levels are lower in paclitaxel-resistant cells [124]. Research indicates that circular RNA (cPKM) can enhance the malignant potential of these cells by promoting the stability of mRNAs such as STMN1, which is closely associated with tumor progression and paclitaxel resistance. cPKM interacts with IGF2BP2 to stabilize STMN1 and TGFB1 mRNA, thereby facilitating the proliferation and metastasis of ovarian cancer cells [116]. This mechanism highlights IGF2BP2’s role as a crucial player in the molecular network that supports cancer cell survival against paclitaxel treatment. Importantly, novel therapeutic strategies involving cPKM-targeted siRNA combined with paclitaxel have shown promise in sensitizing resistant cells, demonstrating the potential of targeting IGF2BP2 to enhance treatment efficacy.

Other chemotherapeutic drugs. ATRA, an active vitamin derivative, has revolutionized the treatment of acute promyelocytic leukemia (APL) by significantly improving patient outcomes [125]. However, resistance to ATRA is a critical challenge, particularly associated with the expression of MN1, a fusion partner of the TEL transcription factor. MN1 promotes both proliferation and self-renewal while inhibiting differentiation, leading to poor prognosis and ATRA resistance in AML patients [126]. Recent studies have shown that IGF2BP2 plays a pivotal role in this context by mediating the m6A modification of MN1 mRNA through the METTL14 pathway [18]. This enhancement of MN1 stability contributes to the development of ATRA resistance not only in APL but also in osteosarcoma [116]. Thus, IGF2BP2 serves as a critical regulator, linking MN1 expression and ATRA resistance, and highlighting its potential as a therapeutic target to overcome this resistance. Moreover, antibiotic chemotherapeutic agents are also crucial in cancer treatment due to their ability to disrupt DNA and induce cell death. Doxorubicin, a first-line chemotherapy agent for breast cancer, faces challenges due to resistance mechanisms, prominently mediated by the ATP-binding cassette transporter ABCB1 [127]. IGF2BP2 has been found to regulate the stability of ABCB1 mRNA in an m6A-dependent manner. Specifically, A1BG-AS1, an RNA molecule that interacts with IGF2BP2, upregulates ABCB1 expression, facilitating doxorubicin resistance in breast cancer cells [51]. This interaction underscores that targeting IGF2BP2 could potentially counteract the sensitivity to doxorubicin.

### 4.2. Targeted Chemotherapy Resistance

IGF2BP2 (insulin-like growth factor 2 mRNA-binding protein 2) plays a crucial role in cancer resistance, particularly in resistance to tyrosine kinase inhibitors (TKIs). Firstly, IGF2BP2 affects TKI sensitivity by regulating mRNA stability and translation. Studies have shown that IGF2BP2 is involved in the METTL3-mediated modification of USP13 mRNA, which enhances USP13 stability. USP13, a deubiquitinase, regulates the stability of autophagy-related protein 5 (ATG5), thereby inducing pro-survival autophagy. This process is closely related to imatinib resistance in gastrointestinal stromal tumors (GISTs) [96]. Secondly, IGF2BP2 significantly impacts TKI resistance through its role in signaling pathways. In radioiodine-refractory papillary thyroid carcinoma, IGF2BP2-dependent ERBB2 signaling activation contributes to resistance against various TKIs [66]. ERBB2 (HER2) is a critical target in many cancer types, and its inhibition is fundamental to several anti-cancer drugs [128]. By enhancing ERBB2 signaling, IGF2BP2 leads to TKI resistance in cancer cells, indicating its important role in regulating signaling pathways and influencing drug responses. Furthermore, IGF2BP2 is involved in resistance formation by modulating the transcription factor forkhead box O1 (FOXO1). FOXO1 is a key transcription factor regulating the cell cycle and apoptosis [129]. IGF2BP2 inhibits PID1 expression through the DANCR/FOXO1 axis, promoting resistance to glioblastoma cells [130]. This process highlights IGF2BP2’s multifaceted regulatory role in cancer, potentially enhancing tumor cell proliferation and survival by downregulating inhibitory factors.

In conclusion, IGF2BP2 significantly affects cancer cell resistance to tyrosine kinase inhibitors by regulating mRNA stability, signaling pathways, and transcription factor activity. Targeting IGF2BP2 is considered a promising strategy to enhance the effectiveness of targeted therapies and may offer new approaches to overcoming cancer resistance.

### 4.3. Immunotherapy

IGF2BP2 has emerged as a critical player in regulating immune therapy resistance in various cancers. Its role in modulating immune escape mechanisms highlights its potential as a therapeutic target, particularly in combination with immune checkpoint inhibitors like CTLA-4 (Figure 4). IGF2BP2 participates in the methylation of circRNA. Circular RNA is associated with immune evasion in CRC [131]. Research reveals that IGF2BP2 can enhance the stability of circQSOX1 through METTL3-mediated N6-methyladenosine (m6A) modification. This circRNA then regulates the miR-326/miR-330-5p/PGAM1 axis, promoting glycolysis and the development of CRC and aiding in immune escape at the same time [117]. Furthermore, IGF2BP2 is implicated in the induction of intratumoral Tregs through its interaction with circQSOX1. The presence of Tregs in the tumor microenvironment is known to suppress anti-tumor immune responses, thereby promoting resistance to immunotherapy [117]. By fostering an environment conducive to Treg accumulation, IGF2BP2 reinforces the resistance of CRC to CTLA-4 blockade, emphasizing its role in shaping the tumor immune landscape.

Considering the significant involvement of IGF2BP2 in immune resistance, targeting this protein presents a promising strategy to enhance the efficacy of immune therapies. By co-targeting IGF2BP2 alongside immune checkpoint inhibitors, it may be possible to acquire better efficacy.

### 4.4. Radiation Therapy Resistance

IGF2BP2 plays a crucial role in the mechanism of radiation resistance. In radiation-resistant lung cancer cells, IGF2BP2 expression is significantly upregulated. This upregulation contributes to radiation resistance through a positive feedback loop with the amino acid transporter SLC7A5, which subsequently activates the Akt/mTOR signaling pathway, promoting cell survival and proliferation. Specifically, IGF2BP2 enhances the stability of SLC7A5 mRNA, increasing SLC7A5 expression and facilitating the transport of methionine (Met) into the cells [118]. Met is essential for the synthesis of S-adenosylmethionine (SAM), which serves as a methyl donor for histone lysine methyltransferase SETD1A. SETD1A catalyzes the trimethylation of histone H3 at lysine 4 (H3K4me3) on the IGF2BP2 gene promoter, thereby boosting IGF2BP2 transcription. This cascade ultimately forms a signaling pathway that supports cell survival, enhancing the resistance of lung cancer cells to radiation therapy. Additionally, IGF2BP2 stability is regulated by the FBW7 ubiquitin ligase, which targets phosphorylated IGF2BP2 for degradation. This regulatory mechanism plays a key role in controlling IGF2BP2 expression levels and function. The FBW7/GSK3β/IGF2BP2/SLC7A5 axis significantly affects cell growth and survival in lung cancer, further increasing resistance to radiation therapy [118]. Dysregulation of IGF2BP2 expression plays a critical regulatory role in tumor radiation resistance, and its dysregulation may serve as an important biomarker for predicting and improving radiation therapy outcomes. Understanding the role of IGF2BP2 in radiation resistance will provide new insights and directions for clinical treatment strategies.

## 5. Targeting the IGF2BP2 to Improve the Effect on Cancer Treatment

The dysregulated role of IGF2BP2 in cancer development [121], progression [114], prognosis [132], and resistance [133] has been established. Increasing evidence points to IGF2BP2 as a potential therapeutic target for cancer. For example, IGF2BP2 was considered as a promising prognostic biomarker and therapeutic target for patients with metastatic clear cell renal cell carcinoma (ccRCC) because it inhibits the invasion and migration of ccRCC cells [134]. Due to the promotion of angiogenesis and metastasis, targeting IGF2BP2 is regarded as a potential antiangiogenic strategy for LUAD [90].

Currently, several studies are exploring small-molecule inhibitors or targeted molecules to inhibit the function of IGF2BP2. Chanda et al. have used CRISPR/Cas9 to knock out the IGF2BP2 gene in lung cancer cell lines (A549, LLC1) and hepatocellular carcinoma cell lines (HepG2, Huh7). They found that IGF2BP2 knockout decreased cell proliferation and migration. In 3D cultures, IGF2BP2 knockout strains transformed from compact spheroids to loose aggregates. And the clonogenic capacity was significantly reduced. Similarly, monoallelic knockout IGF2BP2 clones of Huh7, HepG2, and SW480 cells showed a significant decrease in cell migration [135]. In T cell acute lymphoblastic leukemia (T-ALL), the overexpression and knockdown of IGF2BP2 proved its essential role in the proliferation of T-ALL cells [136]. These studies demonstrate its potential as an anti-cancer target. Moreover, targeting IGF2BP2 has also shown potential in addressing resistance in cancer treatment. In papillary thyroid carcinoma (PTC), knocking down IGF2BP2 inhibited ERBB2 signaling in PTC cells and partially reversed the sunitinib resistance induced by TKI, suggesting that targeting IGF2BP2 is a strategy for reversing acquired TKI resistance [66]. IGF2BP2 is highly expressed in triple-negative breast cancer (TNBC). IGF2BP2 can directly regulate CDK6 m6A-dependent translation, promoting the G1/S phase transition of the cell cycle. Thus, downregulating IGF2BP2 can enhance the sensitivity to CDK4/6 inhibitors as a treatment strategy [85]. It was also found that IGF2BP2 knockdown not only significantly inhibited the proliferation and invasion of papillary thyroid carcinoma (PTC) cells in vitro but also increased the sensitivity of PTC cells to cisplatin treatment [137].

Compounds belonging to benzamide–benzoic acids and urea–thiophenes have been identified to inhibit the interaction between IGF2BP2 and its target RNA. Several compounds, belonging to either benzamidobenzoic acid or ureidothiophene, have been found to selectively restrain the interaction of IGF2BP2 with its target RNAs. In 2022, Charlotte’s team discovered a class of IGF2BP2 inhibitors that utilized fluorescence polarization (FP) to screen for ten compounds [138]. They identified ten compounds that effectively inhibit the binding of IGF2BP2 to RNA-A and RNA-B sequences. These inhibitors have shown particular promise in treating cancers that exhibit elevated IGF2BP2 expression, such as CRC and HCC. In vitro assays, including MTT assays, revealed that compounds 8 and 9 exhibited potent activity in IGF2BP2-expressing cancer cells^32^. Notably, compound 8 also demonstrated significant anti-cancer effects in zebrafish embryos. Structural studies using saturation transfer difference nuclear magnetic resonance (STD-NMR) and thermal shift assays confirmed that these compounds bind to IGF2BP2, with compound 8 showing strong binding to the KH3-4 domain [138]. Chanda S and others tested three of these compounds for their effects on lung cancer, liver cancer, and colorectal cancer cell lines. Three IGF2BP2 inhibitors were able to inhibit cancer cell colony formation and migration. This further confirms the efficacy of IGF2BP2 inhibitors in cancer therapy and lays the foundation for their broader clinical application [135]. However, further clinical trials are needed to fully assess their safety and potential side effects.

Feng and colleagues conducted virtual high-throughput screening targeting the KH3-4 domain of IGF2BP2, leading to the identification of compound 11 (JX5). This compound was optimized through internal fluorescence quenching and Cell Counting Kit-8 (CCK8) assays, showing efficacy against T cell acute lymphoblastic leukemia (T-ALL) [136]. Compound 11 demonstrated a Kd value of 93.2 ± 3.9 μM in binding assays and effectively inhibited T-ALL progression in mouse models. The safety profile of JX5 appears promising, though more detailed studies are necessary to fully understand its toxicity and potential side effects. JX5 inhibits IGF2BP2 by binding to the KH3-4 domain, thereby preventing its interaction with target RNAs involved in leukemia progression. This disruption impedes the survival and proliferation of cancerous T cells [136].

Hydroxybenzyl derivatives have emerged as promising IGF2BP2 inhibitors through structure-based virtual screening conducted by Wang and colleagues. Compound 12 (CWI1-2) was identified as a potent anti-leukemic agent [99]. Compound 12 showed competitive inhibition of RNA binding, with IC50 values ranging from 200 to 700 nM in various leukemia cell lines. It effectively reduced ATP production in acute myeloid leukemia (AML) cells and demonstrated anti-tumor effects in mouse models. The safety profile is favorable, but additional studies are required to confirm its clinical potential and safety [99]. CWI1-2 inhibits IGF2BP2 by competing with RNA for binding, thereby interfering with the stabilization of mRNAs crucial for leukemia cell survival. This mechanism leads to reduced proliferation of leukemia stem cells and improved treatment outcomes [99].

These inhibitors, including ten compounds of benzamidobenzoic acid class and ureidothiophene class, JX5 and CWI1-2 have shown promising results in preclinical studies. They effectively block the interaction between IGF2BP2 and its target RNAs, leading to reduced cancer cell viability, which demonstrates its anti-tumor potential. However, translating these inhibitors into clinical practice remains challenging. Further clinical trials are necessary to evaluate their safety, efficacy, and potential use in various cancer types. In summary, we still hold promise for targeting IGF2BP2 to enhance the efficacy of cancer treatment and potentially overcoming resistance to conventional therapies.

## 6. Conclusions

This article discusses the structure and physiological functions of IGF2BP2, highlighting its role in cancer development and progression, as well as its involvement in tumor drug resistance. IGF2BP2 is a promising therapeutic target in cancer treatment due to its extensive role in promoting oncogenic processes such as tumor growth, metastasis, drug resistance, and metabolic reprogramming. As an m6A reader, IGF2BP2 stabilizes m6A-modified RNAs, influencing gene expression in ways that support cancer cell stemness, proliferation, migration, and evasion of programmed cell death. Additionally, IGF2BP2’s involvement in metabolic reprogramming, specifically in glycolysis and amino acid metabolism, aids cancer cells in adapting to metabolic stress, thus promoting tumor growth. In the TME, IGF2BP2 fosters immunosuppressive conditions by supporting TAMs and other immunosuppressive cells, which help cancer cells escape immune detection. IGF2BP2 also regulates ncRNAs and circRNAs, which contribute to tumor initiation and progression, and promotes key metastatic processes like angiogenesis and EMT. By targeting IGF2BP2, it may be possible to disrupt these cancer-promoting pathways, enhancing the effectiveness of chemotherapy, immunotherapy, and metabolic inhibitors. This comprehensive targeting approach holds potential for slowing cancer progression, reducing metastatic risk, overcoming therapeutic resistance, and improving overall patient outcomes. A deeper understanding of IGF2BP2’s mechanisms in cancer could offer new perspectives and strategies for achieving optimal treatment outcomes.

Expanding on this, IGF2BP2’s ability to bind to and stabilize mRNAs makes it a critical player in the post-transcriptional regulation of gene expression. This regulation is crucial in cancer, where IGF2BP2 can influence various aspects such as cell proliferation, apoptosis, and the cellular response to therapy. By modulating m6A modifications on target mRNAs, IGF2BP2 affects key signaling pathways that drive tumorigenesis and resistance to conventional treatments, including chemotherapy, targeted therapies, and radiotherapy.

Given the multifaceted role of IGF2BP2 in cancer, targeting this protein could disrupt the pathological processes that support tumor growth and survival. Furthermore, its expression levels and activity could serve as biomarkers to predict patient response to certain therapies, enabling a more personalized approach to cancer treatment. Research into IGF2BP2’s interactions and regulatory networks is crucial, as it may reveal additional targets for intervention, ultimately leading to improved therapeutic strategies and outcomes for patients with cancer. Understanding these mechanisms at a molecular level could revolutionize current treatment paradigms and lead to the development of more effective and durable cancer therapies.

## Figures and Tables

**Figure 1 ijms-25-12150-f001:**
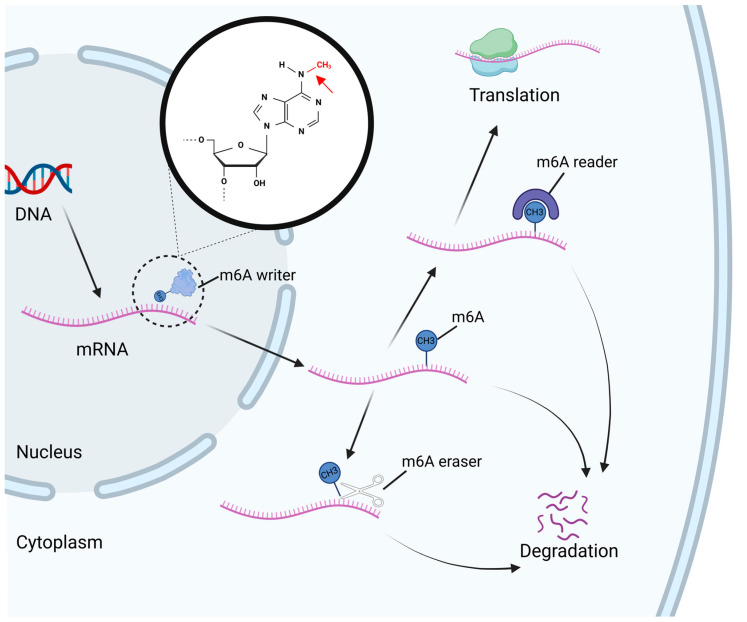
The process of m6A modification. After transcription, the N6 position of adenosine in mRNA can be modified by the addition of a CH3 group as the function of m6A writer, resulting in m6A modification. However, this modification is not stable, and the mRNA may be subject to degradation. Alternatively, specific m6A erasers can remove the methyl group, also leading to mRNA degradation. Conversely, if an m6A reader protein binds to the modified site, the m6A modification is stabilized, allowing the mRNA to be successfully translated into protein as well as degradation.

**Figure 2 ijms-25-12150-f002:**
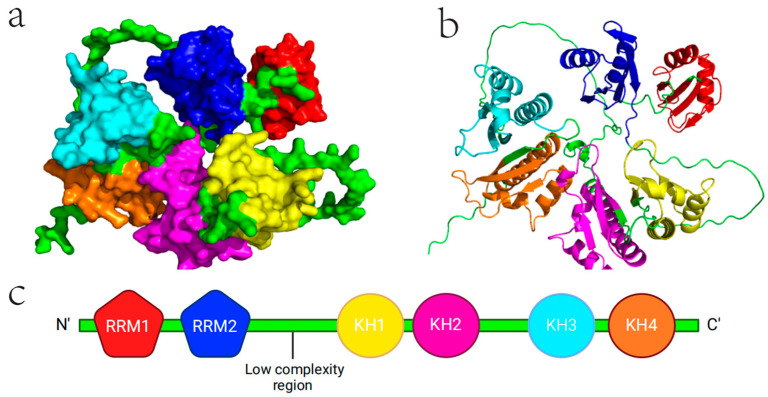
Protein structure of IGF2BP2. (**a**,**b**) are 3D protein structure of IGF2BP2 while (**c**) is unfolding chain. Different colors represent different domains. RRM1 is represented by red, RRM2 is represented by blue, KH1 is represented by yellow, KH21 is represented by magenta, KH3 is represented by cyan, KH4 is represented by orange, and the remaining low-complexity region is represented by green.

**Figure 3 ijms-25-12150-f003:**
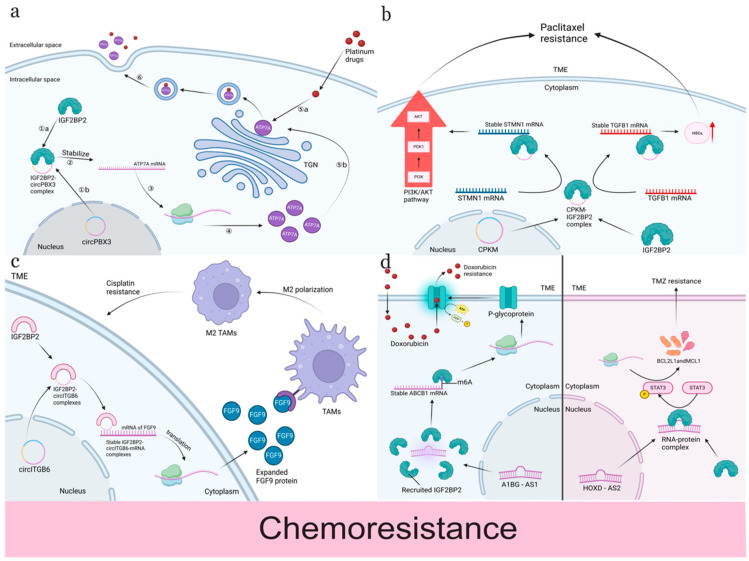
The mechanisms of IGF2BP2 regulating chemotherapy resistance. (**a**) circPBX3 can interact with the RNA-binding protein IGF2BP2 in the cytoplasm to form a complex. The complex increases the stability and translation of ATP7A mRNA and upregulates ATP7A protein levels. ATP7A is stored in the trans-Golgi (TGN). When platinum-based drugs enter cancer cells, ATP7A can bind and segregate them into the vesicles. Vesicles carrying platinum drugs will be translocated outside the cell, leading to drug resistance. (**b**) cPKM is a newly identified circRNA. It binds to IGF2BP2 and STMN1/TGFB1 mRNA 3′ UTR to form an RNA–protein complex, which promotes the interaction of IGF2BP2 with STMN1/TGFB1 mRNA and thus enhances the stability of STMN1/TGFB1 mRNA. TGFB1 is key to the transformation of activated HSCs to a myofibroblast-like phenotype. TGFB1 secreted by tumor cells in the hepatic microenvironment induces the activation of HSCs to produce many extracellular matrix components, which shapes the microenvironment for tumor progression, leading to the decrease in paclitaxel sensitivity in cancer cells. STMN1 reduces the sensitivity to paclitaxel of cancer cells by activating the PI3K/Akt pathway. (**c**) IGF2BP2 binds to the CAUC motif of circITGB6 in the cytoplasm via the KH1-2 bi-structural domain. Two CAAAC sites within circITGB6 can directly bind to AU-element-rich FGF9, leading to the formation of the circITGB6/IGF2BP2/FGF9 ternary complex. The formation of the complex enhances the stability of FGF9 mRNA, which leads to increased extracellular FGF9 secretion. This ultimately promotes an increase in FGF9 in the TME of ovarian cancer cells, induces macrophage polarization toward the M2 phenotype, and ultimately leads to platinum resistance in OC patients. (**d**) In the cytoplasm, A1BG–AS1 recruits IGF2BP2, and the ABCB1 3′ UTR has a very-high-confidence m6A modification site that is bound by IGF2BP2, preventing degradation and stabilizing the expression of ABCB1. ABCB-1 encodes P-gp, a protein located in the cell membrane, which is capable of binding to doxorubicin and utilizing the hydrolysis of ATP to provide energy for drug transport. HOXD-AS2 is an IncRNA that interacts with IGF2BP2 to form an RNA–protein complex to fulfill its molecular function. Upregulated HOXD–AS2/IGF2BP2 promotes STAT3 signaling, phosphorylates tyrosine 705 residue of STAT3, and upregulates the expression of BCL2L1 and MCL1, which promotes anti-apoptotic activity and reduces the responsiveness of cancer cells to chemotherapeutic drug TMZ.

**Figure 4 ijms-25-12150-f004:**
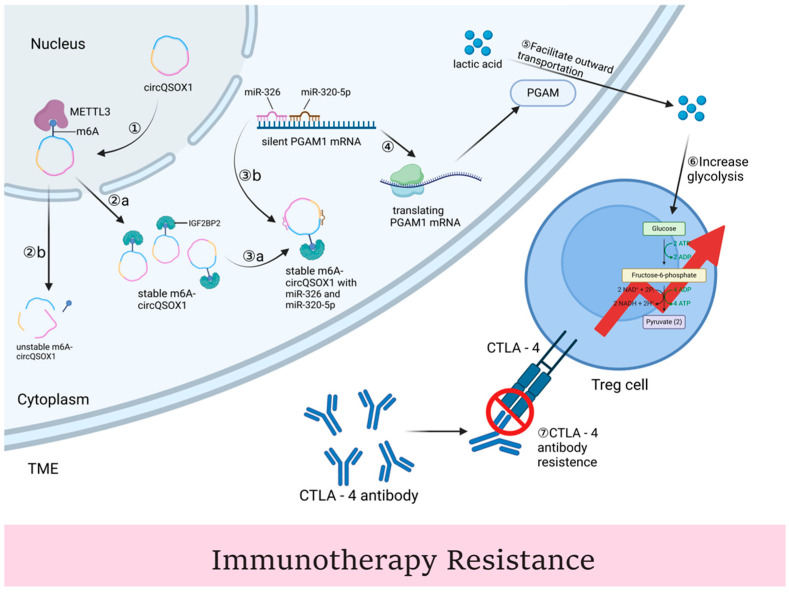
The mechanisms of IGF2BP2 regulating immunotherapy resistance. ① METTL3 modifies circQSOX1 with m6A, while ②a IGF2BP2 binds to the m6A modification on cicrQSOX1, stabilizing and upregulating cicrQSOX1, or the ②b unstable cicrQSOX1 would be degraded. miR-326 and miR330-5p can bind to cicrQSOX1 to ④ release PGAM1 RNA for translation. Meanwhile, whereas cicrQSOX1 adsorbs miR-326 and miR-330-5p, PGAM1 is upregulated, which ⑤ promotes lactate translocation to the extracellular compartment, ⑥ increases cytosolic glycolytic activity, and facilitates Treg cell-associated CRC immune escape, therefore ⑦ resisting the anti-cancer effects of the CTLA-14 antibody.

**Table 1 ijms-25-12150-t001:** m6A interactions across cell types in cancer contexts.

Cell Type	Normal Function of m6A	Oncogenic Function of m6A	Reference
Stem Cells	Regulates self-renewal and differentiation.	Promotes stemness and tumorigenesis.	[103]
T Cells	Modulates activation and differentiation.	Inhibits anti-tumor immunity by enhancing T cell exhaustion.	[104,105]
Cancer Cells	Maintains normal cell growth and apoptosis.	Promotes proliferation and survival by altering metabolism.	[106,107]
Macrophages	Regulates inflammatory response and tissue repair.	Enhances M2 polarization, supporting tumor progression.	[108]
Endothelial Cells	Facilitates angiogenesis in normal tissue development.	Drives abnormal angiogenesis in tumors.	[109]

**Table 2 ijms-25-12150-t002:** The role of IGF2BP2 in various cancer treatments.

Cancer Treatment	Drug	Cancer Type	Effect	Mechanism	Ref
Chemotherapy	Cisplatin	Ovarian cancer	Induce resistance	circRNA interacts with IGF2BP2-FGF9 complex to induce polarization of TAMs toward M2 phenotype to enhance CDDP resistance.	[111]
	Cisplatin	Ovarian cancer	Induce resistance	circPBX3/IGF2BP2/ATP7A axis to induce efflux of cisplatin.	[111]
	Cisplatin	Cervical cancer	Induce resistance	miR-96-5p enhances IGF2BP2 to induce drug resistance.	[112]
	Oxaliplatin	Cervical cancer	Induce resistance	WTAP/IGF2BP2/lnc-OXAR axis regulates the recruitment of ku70 and cystatin A to facilitate DNA double-strand break repair, which enhances drug resistance.	[113]
	TMZ	Glioblastoma multiforme	Induce resistance	IGF2BP2/IGF2/PI3K/Akt signaling pathway to regulate TMZ resistance.	[114]
	TMZ	Glioma cells	Induce resistance	SOX2/IGF2BP2/DHX9 axis to increase TMZ resistance.	[115]
	TMZ	Glioblastoma multiforme	Induce resistance	HOXD-AS2/IGF2BP2/STAT3 positive feedback loop to regulate the sensitivity of TMZ.	[28]
	Paclitaxel	Ovarian cancer cells	Induce resistance	cPKM-IGF2BP2/STMN1/TGFB1 axis to facilitate proliferation and metastasis and increase the drug resistance.	[116]
	ATRA	Acute promyelocytic leukemia	Induce resistance	METTL14/IGF2BP2/MN1 axis leads to ATRA resistance.	[18]
	Doxorubicin	Breast cancer cells	Induce resistance	A1BG-AS1/IGF2BP2/ABCB1 axis.	[51]
Targeted chemotherapy	Imatinib	Gastrointestinal stromal tumors	Induce resistance	METTL3/IGF2BP2/USP13/ATG5 axis.	[96]
Immunotherapy	Immune checkpoint like CTLA-4	Colorectal cancer	Induce resistance	miR-326/miR-330-5p/PGAM1 axis AND induce Tregs.	[117]
Radiation therapy	Radiation	Lung cancer	Induce resistance	FBW7/GSK3β/IGF2BP2/SLC7A5 axis.	[118]

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
