# Peer review of "The Emerging Role of IGF2BP2 in Cancer Therapy Resistance: From Molecular Mechanism to Future Potential"

_ijms, 2024, doi:10.3390/ijms252212150_

Round 1

Reviewer 1 Report

Comments and Suggestions for Authors

Reviewer: While reviewing the manuscript “The emerging role of IGF2BP2 in cancer therapy resistance: From molecular mechanism to future potential” This manuscript discusses the structure and physiological function of IGF2BP2, with emphasis on its role in cancer development and progression, as well as its involvement in cancer drug resistance. As a m6A-modified reading protein, IGF2BP2 impacts cancer therapy by modulating m6A modification. However, I would like to mention several points that I feel should be addressed before I could recommend this manuscript for publication.

Comment 1: It is recommended to investigate in depth how IGF2BP2 regulates m6A modification to affect cancer development, progression and drug resistance, and its specific mechanism of action in tumors.

Comment 2: Given the critical role of IGF2BP2 in cancer, could drugs that can inhibit or modulate IGF2BP2 function be developed as novel therapeutic approaches to intervene tumor growth and drug resistance.

Comment 3: In addition to the currently investigated colorectal cancer, hepatocellular carcinoma and leukemia, it is recommended to evaluate the efficacy of IGF2BP2 inhibitors in other cancers with high IGF2BP2 expression, such as breast cancer and lung cancer, and expand their scope of application.

Comment 4: Figure 1 is missing m6A writers, which makes the picture not form a closed loop. In addition, different reading proteins have different functions, not all reading proteins can promote RNA stability, some reading proteins can also promote RNA degradation, such as YTHDF2. Therefore, the x in the picture is not valid, please ask the author to correct it.

Comment 5: Different colours in Figure 2 indicate different structural domains, please explain what the green colour in Figure arepresents.

Comment 6: The clarity of Figure S1 is too low, please improve the clarity of the picture.

Author Response

Thank you very much for your careful review, here are our response.

Comment 1: It is recommended to investigate in depth how IGF2BP2 regulates m6A modification to affect cancer development, progression and drug resistance, and its specific mechanism of action in tumors.

Response 1: Agree. we have investigated in depth how IGF2BP2 regulates m6A modification to affect cancer development, progression and drug resistance. “IGF2BP2 plays a critical role in cancer development, progression, and drug resistance by modulating m6A RNA modifications. As a reversible and prevalent modification, m6A occurs on RNA molecules (including mRNA and ncRNA) through methylation at the nitrogen-6 position of adenosine within an RRACH sequence (where R = G or A, and H = A, U, or C). This modification process is coordinated by three types of proteins: writers (methyltransferases), readers (m6A-binding proteins), and erasers (demethyl-ases). Writers catalyze the addition of methyl groups to target RNAs, while erasers remove these methyl groups, enabling dynamic regulation of m6A levels. Readers recognize m6A-marked binding sites on RNA, which recruits writers and affects RNA folding, stability, degradation, and cellular interactions. Through these mechanisms, m6A modifications are involved in RNA splicing, translation, export, and decay. In cancer, IGF2BP2, a common m6A reader, exhibits aberrant expression, impacting the stability and expression of various mRNAs associated with oncogenes (e.g., MYC, KRAS, BCL2), metabolism-related genes (e.g., GLUT1, HK2), stemness-related genes (e.g., SOX2, NANOG, OCT4), and drug-resistance genes (e.g., ABCB1, which encodes multidrug resistance proteins). Consequently, IGF2BP2 significantly influences tumor growth, progression, and resistance to therapies.” (Page 6, Line 227-238)

Comment 2: Given the critical role of IGF2BP2 in cancer, could drugs that can inhibit or modulate IGF2BP2 function be developed as novel therapeutic approaches to intervene tumor growth and drug resistance.

Response 2: Thank you for the suggestion.I think this advice is very good. I can make the article more logical and also explain why we are introducing IGF2BP2 inhibitors. we have provided some examples to describe the vital role of IGF2BP2 in cancer growth,metastasis and provide some evidence to prove that inhibit or modulate IGF2BP2 function can be developed to regulate the cancer treatment and drug resistance. (Page 18,Line 719-739).

Comment 3: In addition to the currently investigated colorectal cancer, hepatocellular carcinoma and leukemia, it is recommended to evaluate the efficacy of IGF2BP2 inhibitors in other cancers with high IGF2BP2 expression, such as breast cancer and lung cancer, and expand their scope of application.

Response 3: Thank you for the suggestion. It is a really good question. I added the study to show the effect of IGF2BP2 inhibitors on the lung cancer.(Page 18, Line 751-755).

Comment 4: Figure 1 is missing m6A writers, which makes the picture not form a closed loop. In addition, different reading proteins have different functions, not all reading proteins can promote RNA stability, some reading proteins can also promote RNA degradation, such as “YTHDF2”. Therefore, the “x” in the picture is not valid, please ask the author to correct it.

Response 4: The missed m6A writers was annotated to form a closed loop, and the “x” was deleted to correct the mistake. (Page 2, Line 59)

Comment 5: Different colours in Figure 2 indicate different structural domains, please explain what the green colour in “Figure a” represents.

Response 5: The green colour structure was remained low complexity region, and was annotated on figure 2 now. (Page 3, Line 101)

Comment 6: The clarity of Figure S1 is too low, please improve the clarity of the picture.

Response 6: The clear Figure S1 was provided now. (Page 7, Line 285)

Reviewer 2 Report

Comments and Suggestions for Authors

This review focuses on the increasingly recognized role of IGF2BP2 in cancer therapy resistance, highlighting its functions as an m6A "reader" in RNA methylation and post-transcriptional regulation. It provides a well-rounded overview, effectively framing IGF2BP2's dual role in tumor progression and drug resistance. The stated aim of the review—enhancing the understanding of IGF2BP2 mechanisms to improve chemotherapy efficacy—underscores its clinical relevance and potential impact on patient outcomes. The manuscript is well-written and presents a thorough analysis. However, I would suggest addressing the following points to enhance clarity and depth:

1.       Could simplifying or restructuring some descriptions improve reader comprehension, especially for complex interactions like the "2+4" modular architecture?

2.       Could you clearly describe the m6A-dependent mechanisms by which IGF2BP2 stabilizes RNA? Would a diagram or table summarizing the m6A interaction across cell types, particularly in cancer contexts, help clarify the differences between its normal and oncogenic functions?

3.       Given IGF2BP2’s involvement in multiple physiological and pathological processes, could the article elaborate on potential therapeutic implications, such as targeting IGF2BP2 for metabolic, renal, or degenerative diseases?

4.       In part2.2, the section suggests IGF2BP2 as a potential biomarker for cancer progression and therapy. Are there specific cancers in which IGF2BP2 has already been linked to prognosis or treatment response? Including examples could strengthen this point.

5.       Given that IGF2BP2 is overexpressed in HPV-negative HNSC and metastatic melanoma, could the section discuss the significance of this finding? For instance, could IGF2BP2 expression levels be involved in HPV-related mechanisms or metastatic processes in melanoma?

6.       Is it known whether IGF2BP2 promotes the translation of other oncogenic targets across different cancers? This information could highlight how IGF2BP2’s translational activity affects cancer progression.

7.       Are there any experimental studies or clinical trials focusing on targeting IGF2BP2 in cancer that could be discussed to underscore the protein’s clinical relevance?

8.       As IGF2BP2 is considered as a potential biomarker, especially regarding treatment resistance, could data be provided that highlight any correlations between IGF2BP2 expression levels and patient prognosis or responsiveness to treatment, particularly in cancers like colorectal or lung cancer?

Author Response

Thank you very much for taking the time to review this manuscript. Please find the detailed responses below and the corresponding revisions highlighted in the re-submitted files.

  1. Could simplifying or restructuring some descriptions improve reader comprehension, especially for complex interactions like the "2+4" modular architecture?

Response 1: Yes, we have simplifying descriptions to improve reader comprehension like "2+4" modular architecture, the details was annotated in the revised manuscript. (Page 3, Line 89-90)

  1. Could you clearly describe the m6A-dependent mechanisms by which IGF2BP2 stabilizes RNA? Would a diagram or table summarizing the m6A interaction across cell types, particularly in cancer contexts, help clarify the differences between its normal and oncogenic functions?

Response 2: We have added some detailed describe about m6A-dependent mechanisms by which IGF2BP2 stabilizes RNA. The context was “1) Splicing and decapping: m6A modification regulates the splicing and decapping of mRNA, thereby promoting the formation of mature mRNA. Proper splicing and decapping can improve the stability of mRNA. 2) Binding Specific Proteins: m6A modifications can recruit specific RNA-binding proteins, such as YTHDF family pro-teins, that protect RNA from degradation or promote its translation by binding to m6A-modified sites. 3) Inhibition of RNA degradation: m6A modification inhibits specific RNA degradation pathways, such as the "deadenylation" pathway, thereby extending the half-life of mRNA. Modified RNA is less likely to be degraded by ribo-nucleases. 4) m6A methylation can improve the translation efficiency of mRNA and enhance protein synthesis, thereby indirectly improving RNA stability, because more translation products will promote the maintenance of RNA presence in cells. 5) m6A can synergize with other RNA modifications, such as m5C, to form a complex regulatory network that affects RNA stability and function.” (Page 4, Line 149-161)

we also added a table that summarize the m6A interaction across cell types in cancer contexts,the detailed table was added to the revised manuscript. (Page 12, Line 529-530)

  1. Given IGF2BP2’s involvement in multiple physiological and pathological processes, could the article elaborate on potential therapeutic implications, such as targeting IGF2BP2 for metabolic, renal, or degenerative diseases?

Response 3: Thank you for your feedback. After careful consideration and discussion, we have decided to focus our article specifically on the role of IGF2BP2 in cancer and cancer drug resistance. Due to space limitations, we are unable to extensively cover its therapeutic significance in diseases other than cancer. However, we have ensured that the therapeutic implications of IGF2BP2 for cancer are integrated into each relevant section of the manuscript, as indicated in the annotations. (Page 20, Line 812-827)

  1. In part2.2, the section suggests IGF2BP2 as a potential biomarker for cancer progression and therapy. Are there specific cancers in which IGF2BP2 has already been linked to prognosis or treatment response? Including examples could strengthen this point.

Response 4: Thank you for your comment. The part about IGF2BP2 as a potential biomarker for cancer progression and therapy was added on part 4, and was write in red colour. (Page18, Line 712-739)

  1. Given that IGF2BP2 is overexpressed in HPV-negative HNSC and metastatic melanoma, could the section discuss the significance of this finding? For instance, could IGF2BP2 expression levels be involved in HPV-related mechanisms or metastatic processes in melanoma?

Response 5: There are no research about IGF2BP2 and HPV-negative HNSC as well as metastatic melanoma, the different expression was solely based on existing data base, and we have added the true research situation to the article. (Page 6, Line 259-260)

  1. Is it known whether IGF2BP2 promotes the translation of other oncogenic targets across different cancers? This information could highlight how IGF2BP2’s translational activity affects cancer progression.

Response 6: IGF2BP2 promotes the translation of other oncogenic targets across different cancers, here are a summary and the detail gene of each cancer was introduced within each part.

“In cancer, IGF2BP2, a common m6A reader, exhibits aberrant expression, impacting the stability and expression of various mRNAs associated with oncogenes (e.g., MYC, KRAS, BCL2), metabolism-related genes (e.g., GLUT1, HK2), stemness-related genes (e.g., SOX2, NANOG, OCT4), and drug-resistance genes (e.g., ABCB1, which encodes multidrug resistance proteins). Consequently, IGF2BP2 significantly influences tumor growth, progression, and resistance to therapies.” (Page 6, Line 239-244)

  1. Are there any experimental studies or clinical trials focusing on targeting IGF2BP2 in cancer that could be discussed to underscore the protein’s clinical relevance?

Response 7: Thank you for your comment. I think it is a great suggestion to provide enough evidence to highlight the clinical significance of IGF2BP2. There are few clinical studies focusing on targeting IGF2BP2 in cancer. But we added enough experiemental studies to underscore the  IGF2BP2’s clincial relevance. (Page18, Line 712-739).

  1. As IGF2BP2 is considered as a potential biomarker, especially regarding treatment resistance, could data be provided that highlight any correlations between IGF2BP2 expression levels and patient prognosis or responsiveness to treatment, particularly in cancers like colorectal or lung cancer?

Response 7: we have provided data that highlight correlations between IGF2BP2 expression levels and patient prognosis and responsiveness to treatment. “A comprehensive pan-cancer analysis has demonstrated that elevated expression levels of IGF2BP2 are associated with poor prognosis across various cancer types. Specifically, high IGF2BP2 expression significantly correlates with unfavorable outcomes in patients with adrenocortical carcinoma (ACC) (HR=1.17, P <0.03), bladder carcinoma (BLCA) (HR=1.09, P <0.01), kidney renal clear cell carcinoma (KIRC) (HR = 1.21, P <0.001), low-grade glioma (LGG) (HR = 1.48, P <0.001), head and neck squamous cell carcinoma (HNSC) (HR=1.17, P <0.001), lung adenocarcinoma (LUAD) (HR = 1.12, P <0.02), acute myeloid leukemia (LAML) (HR = 1.10, P <0.03), liver hepatocellular carcinoma (LIHC) (HR = 1.11, P <0.01), pancreatic adenocarcinoma (PAAD) (HR = 1.45, P <0.001), and mesothelioma (MESO) (HR = 1.27, P <0.001). These findings suggest that IGF2BP2 may act as an oncogene, contributing to adverse prognoses in multiple cancer types [38]. In various preclinical models of colorectal cancer (CRC), IGF2BP2 was identified as the most prevalent IGF2BP family member in both primary and metastatic CRC, showing a correlation with tumor stage in patient samples and promoting tumor growth [39]. Furthermore, elevated IGF2BP2 expression in primary tumor tissue was significantly linked to resistance against multiple therapies, including selumetinib, gefitinib, and regorafenib in patient-derived organoids (PDOs), as well as 5-fluorouracil and oxali-platin [39]. Similar associations have been observed in other cancer types, such as breast cancer [40], lung cancer [41], ovarian cancer [42], GBM [43] ect,. Notably, IGF2BP2 may also play as protect role in cancer like clear cell renal cell carcinoma (ccRCC) [44]. As previously noted, IGF2BP2 functions as an m6A RNA reader, primar-ily exerting its effects through RNA modification. In the following sections, we will review the role of IGF2BP2 in modulating tumor RNA. “ (Page 6-7, Line 263-284)